# The Design, Synthesis, and Evaluation of Diaminopimelic Acid Derivatives as Potential *dap*F Inhibitors Preventing Lysine Biosynthesis for Antibacterial Activity

**DOI:** 10.3390/antibiotics12010047

**Published:** 2022-12-28

**Authors:** Mohd Sayeed Shaikh, Mayura A. Kale, V. Muralidharan, T. Venkatachalam, Syed Sarfaraz Ali, Fahadul Islam, Sharuk L. Khan, Falak A. Siddiqui, Humaira Urmee, Ganesh G. Tapadiya, Sachin A. Dhawale, Long Chiau Ming, Ibrahim Abdel Aziz Ibrahim, Abdullah R. Alzahrani, Md. Moklesur Rahman Sarker, Mohd Fahami Nur Azlina

**Affiliations:** 1Y. B. Chavan College of Pharmacy, Dr. Rafiq Zakaria Campus, Aurangabad 431001, Maharashtra, India; 2Government College of Pharmacy, Aurangabad 431005, Maharashtra, India; 3Vishnu Institute of Pharmaceutical Education and Research, Hyderabad 502313, India; 4JKKMMRFs-Amnai JKK Sampoorani Ammal College of Pharmacy, Erirmedu, Kumarapalaiyam 638183, Tamil Nadu, India; 5Sub District Hospital, Ambad, Dist. Jalna, Maharashtra 431204, India; 6Department of Pharmacy, Faculty of Allied Health Sciences, Daffodil International University, Dhaka 1207, Bangladesh; 7Department of Pharmaceutical Chemistry, N.B.S. Institute of Pharmacy, Ausa 413520, Maharashtra, India; 8Department of Pharmaceutical Science, North South University, Dhaka 1229, Bangladesh; 9Shreeyash Institute of Pharmaceutical Education and Research, Aurangabad 431005, Maharashtra, India; 10School of Medical and Life Sciences, Sunway University, Sunway City 47500, Malaysia; 11Department of Pharmacology and Toxicology, Faculty of Medicine, Umm Al-Qura University, Makkah 24382, Saudi Arabia; 12Department of Pharmacy, State University of Bangladesh, 77 Satmasjid Road, Dhanmondi, Dhaka 1205, Bangladesh; 13Health Med Science Research Network, 3/1, Block F, Lalmatia, Dhaka 1207, Bangladesh; 14Department of Pharmacology, Faculty of Medicine, University Kebangsaan Malaysia, Jalan Yacob Latif, Kuala Lumpur 56000, Malaysia

**Keywords:** diaminopimelic acid, *dap*F inhibitors, structure-based drug design, heterocyclic, antibacterial, enzyme

## Abstract

We created thiazole and oxazole analogues of diaminopimelic acid (DAP) by replacing its carboxyl groups and substituting sulphur for the central carbon atom. Toxicity, ADME, molecular docking, and in vitro antimicrobial studies of the synthesized compounds were carried out. These compounds displayed significant antibacterial efficacy, with MICs of 70–80 µg/mL against all tested bacteria. Comparative values of the MIC, MBC, and ZOI of the synthesized compound were noticed when compared with ciprofloxacin. At 200 µg/mL, thio-DAP (1) had a ZOI of 22.67 ± 0.58, while ciprofloxacin had a ZOI of 23.67 ± 0.58. To synthesize thio-DAP (1) and oxa-DAP (2), l-cysteine was used as a precursor for the L-stereocenter (l-cysteine), which is recognized by the *dap*F enzyme’s active site and selectively binds to the ligand’s L-stereocenter. Docking studies of these compounds were carried out using the programme version 11.5 Schrodinger to reveal the hydrophobic and hydrophilic properties of these complexes. The docking scores of compounds one and two were −9.823 and −10.098 kcal/mol, respectively, as compared with LL-DAP (−9.426 kcal/mol.). This suggests that compounds one and two interact more precisely with *dap*F than LL-DAP. Chemicals one and two were synthesized via the SBDD (structure-based drug design) approach and these act as inhibitors of the *dap*F in the lysine pathway of bacterial cell wall synthesis.

## 1. Introduction

Pathogenic bacteria are one of the prime causes of human death and disease. Infections such as typhoid, leprosy, tuberculosis, syphilis, tetanus, diphtheria, cholera, and many others, are caused by bacteria [1,2]. Each pathogen species has a unique set of interactions with its human host. Bacteria, such as Streptococcus and Staphylococcus, cause meningitis, pneumonia, skin infections, and even sepsis, a fatal systemic inflammatory reaction that results in shock and severe vasodilation [3,4,5]. Species such as *Chlamydia* cause urinary tract infections and pneumonia and may also be involved in the pathogenesis of coronary heart disease [6]. A large number of existing antibiotics function by inhibiting the biosynthesis of peptidoglycan in the bacterial cell wall. Many antibiotics target the metabolism of D-amino acids since the human body does not utilize them. Alaphosphin and D-cycloserine, for example, inhibit the biosynthesis of D-alanine, whereas vancomycin binds to peptidyl-D-Ala-D-Ala residues, preventing cross-linking. Phosphonomycin inhibits the enzyme UDP-N-acetylglucosamine-3-enolpyruvyltransferase, whereas bacitracin, tunicamycin, and β-lactam antibiotics inhibit peptidoglycan translocation, thereby inhibiting bacterial cell wall synthesis [1,6,7]. There are relatively few antibiotics with structures related to those of DAP, and of these, none have any therapeutic utility. However, in recent years, there has been a rapid emergence of bacterial resistance to both conventional and more potent antibiotics such as vancomycin [7]. In this present research work, we attempt to target the DAP pathway that functions in the biosynthesis of lysine in bacteria. This could possibly be an approach for the development of antibacterial agents that could combat resistant bacterial strains such as VRSA, MRSA, and VRE [8]. Only a few antibiotics, such as daptomycin and linezolid, have been approved in recent years to combat the threat of drug resistance. VanX is one of five genes that, when activated in pathogenic VRE, cause the production of the cell wall to create precursors of peptidoglycan chains that end in D-alanyl-D-lactate (D-Ala-D-lactate) rather than the typical D-Ala-D-Ala residues [9,10,11,12,13].

The N-acetylglucosamine (NAG) and N-acetylmuramic acid (NAM) sugar units comprise the β-1,4-linked polysaccharide that, in turn, creates the peptidoglycan layer, which forms the bacterial cell wall. A pentapeptide side chain (composed of muramyl residues) with a general biological structure is joined to the lactyl side chain of the NAM unit. L-Lysine, or meso-DAP, is the X in the formula L-Ala-g-D-Glu-X-D-Ala-D-Ala (Figure 1) [1,6,7]. Thus, compounds that inhibit DAP or lysine biosynthesis could be very effective antibiotics when used to inhibit bacterial cell wall biosynthesis. Because mammals lack the DAP biosynthetic pathway and obtain L-lysine from dietary sources, molecules that inhibit DAP-metabolizing enzymes may have low mammalian toxicity [1,6,7,14,15]. Many peptidoglycan monomers, including the potent toxin from *B. pertussis* and *N. gonorrhoeae*, and similar DAP-containing peptides, have antitumor, cytotoxic, immunostimulant, angiotensin-converting enzyme (ACE) inhibitory, and sleep-inducing activities in humans [14,15,16,17].

## 2. Materials and Methods

The thiazole ring is important because it is found in biochemical structures such as thiamine, a coenzyme required for the oxidative decarboxylation of α-keto acids, tetrahydro-thiazole, and the penicillin ring [18]. The most important method of thiazole ring synthesis is Hantzsch thiazole synthesis from thioamides and α-Halocarbonyl compounds. An example of Hantzsch synthesis is the preparation of nizatidine, an anti-histaminic agent [19]. In Hantzsch thiazole synthesis (and its modifications), the condensation of a suitably substituted α-haloketone (3) (or its equivalents) with thioamide (4) has proven to be a good method for the synthesis of 2,4,5-methyl thiazoles. Here, we utilized the procedure of Hantzsch thiazole synthesis for the preparation of 2-methylthiazoles (5) for the synthesis of DAP analogues with the help of thioacetamide (3) and 2-chloroacetaldehyde (4) by thionation-cyclization. The thionation-cyclization of enamide monothioamide intermediates (5a and 5b), yielding (5) after intramolecular (2 + 3) cyclization and dehydration, is demonstrated in Figure 1.

Natural molecules with an oxazole moiety are thought to be derived from amino acids such as serine and threonine. The Robinson-Gabriel-type cyclization includes side-chain oxidation followed by the subsequent cyclodehydration of amide alcohols [20]. The most common method for producing oxazoline is to first prepare a β-hydroxy amide and then cyclize it. Typical reagents for cyclization include Burges’s reagent, PPh3/DIAD, DIC/Cu(OTf)_2_, molybdenum oxide and DAST/Deoxo-Fluor [21]. Currently, the main synthetic methods for the preparation of oxazoles are the cyclodehydration of carbonyl ethoxyamine (11a and 11b) and an intramolecular rearrangement to form with 11, but this reaction cannot occur in the absence of promoters (weak Brønsted and Lewis acids). This is known as the Blümlein-Lewy oxazole synthesis [22]. 4,5-Dihydrooxazole (11b) is used as an intermediate to oxidize an oxazolidine formed when amide condenses with an α-haloaldehyde. The first intermediate is formed as carbonyl ethoxyamine (11a) by the condensation of acyclic acetamide and 2-bromoacetaldehyde, which after an intermolecular rearrangement, produces the cyclic 4,5-dihydrooxazole intermediate (11b), which in turn on hydrolysis provides 11.

The Wohl–Ziegler reaction follows a free-radical route. Bromination is usually performed by using NBS, which is an effective brominating reagent with a focus on the allylic and benzylic positions [23]. This reagent is used to bromate the α-position of the carbonyl group as well as the triple bond. When a compound contains both a double and a triple bond, the bromination should be placed α- to the triple bond. This Wohl-Ziegler bromination property was used in our brominating method to introduce heteroatoms into methyl-substituted thiazole, (5) and oxazole (11), yielding the expected monobromides 7 and 12, as shown in Figure 2 and Figure 3. α-bromination is the first step in inserting a heteroatom into a molecular structure to create a probe for further elongation. According to reports, NBS could be used to perform the α-bromination of carbonyl compounds in the presence of radical initiators such as azobisisobutyronitrile (AIBN) and dibenzoyl peroxide (BPO). According to Das et al. and Tanemura et al., α-bromination with NBS over silica-supported sodium hydrogen sulphate and ammonium acetate yield a high percentage of the corresponding ketones [24,25]. The bromination of the oxazole and thiazole at the 2-methyl positions provides a site for the elongation of the designed DAP analogs. The mono-brominated products 7 and 12 were formed by refluxing 5 and 11 with NBS in the presence of benzoyl peroxide in the CCl_4_ solvent as the free radical initiator. NBS acts as a steady and continuous supply source of bromine in this reaction, which most likely involves free radical bromination with benzoyl peroxide acting as the free radical initiator. In organic chemistry, K_2_CO_3_ is a frequently used base that is used to deprotonate mildly acidic protons, such as those in alcohols, phenols, 1,3-dicarbonyl compounds, and thiols. Relative to alcohols, thiols are more acidic [26]. Dilip Konwar et al. performed the selective oxidation of alcohols in water to aldehydes and ketones [27]. A quick, effective, and high-yield process for oxidizing bromo-methylated 7 and 12 to the corresponding thio-DAP 1 (Figure 2) and oxa-DAP 2 (Figure 2) by reacting them with L-lysine was developed. The stereospecificity of thio-DAP and oxa-DAP in the L-terminus is required for substrate binding, which is our target. This criterion restricts us from freezing one terminal as the L-stereocenter (L-cysteine), which is recognized by the enzyme active site and selectively binds to the L-stereocenter of the ligand.

## 3. Results and Discussion

The structure-based drug design (SBDD) technique was employed for studying the interactions between the already existing analogues of DAP and the enzyme’s active site. Further, SBDD was utilized for designing the new analogues, thio-DAP and oxa-DAP (one and two). The SBDD approach increased the inhibitor binding affinity and cellular potency towards the target site. However, very few investigations on the target characteristics of this amino acid in the development of antibacterial drugs is reported. Inhibitors against the enzyme *dap*F of the lysine biosynthetic pathway were developed. In the present work, compounds one and two were developed in accordance with the reported analogues’ structure-activity relationships and studies of the active sites of the enzymes (Figure 2) [16,28]. In addition, structural DAP analogues with carboxyl groups replaced with heterocycle moieties, such as thiazole and oxazole, as well as the introduction of a sulphur moiety at 2–3 carbon lengths from the terminal heterocycle, were designed and synthesized. These analogues underwent testing as *dap*F enzyme inhibitors and, consequently, as antibacterial agents. Because of their potential biological effects as antibacterial, herbicidal, antitubercular, fungicidal, anti HIV, pesticidal, antiinflammatory, antiprotozoal, antihypertensive, and schizophrenia medications, medicinal chemists are increasingly interested in designing oxazole and thiazole heterocyclic analogues [18,29]. These heterocyclic analogues mimic the important aspects of protein structures or their functions and thus act as bioactive molecules. These heterocycles contain oxygen, nitrogen, or sulphur atoms, which provide interactions with the target site like enzymes, receptors and nucleic acids via forces, such as hydrogen bonding, hydrophobic interactions, weak Vander Waals forces, and ionic bonding, as in the case of captopril binding with ACE [30]. These substances were designed and synthesized to act as reversible or irreversible inhibitors of lysine pathway enzymes [16,31]. These DAP analogues might most likely be changed into analogues of tight-binding transition states at the *dap*F active site [14]. Because these heterocycles act as an important facet of protein structures and provide interactions with the target site or enzymes, they exhibit excellent competitive *dap*F inhibitory activity [16,22]. The synthesized analogues did not exhibit any of the examined toxicity parameters in tests of toxicity risk assessments utilizing the protox-II property explorer. These molecules may therefore serve as good potential candidates for novel antibacterial compounds.

### 3.1. In Silico Toxicity Studies

Here, we used the free and open-source tool ProTox-II Toxicity Explorer to conduct an in silico toxicity risk investigation of compounds one and two. The values obtained were compared with those of the native substrate, viz., LL-DAP (Table 1) [32]. Both the designed compounds (one and two) were found to be nontoxic. This renders them good candidates for future development as bacterial agents. Via this same study, we concluded that our designed molecules are inactive at nuclear receptors such as the androgen receptor (AR), aromatase receptor, and estrogen receptor alpha (ER). ProTox-II also provided the predicted LD_50_ values, and it was found that both our compounds had sufficiently high LD_50_ scores, which indicates their safe and nontoxic nature, as shown in Table 1.

### 3.2. ADME Parameter’s Estimation

The SwissADME programme (http://www.swissadme.ch) (accessed on 1 July 2022) was used to estimate the in silico ADME parameters of the synthesized compounds one, two, and LL-DAP in order to screen for their pharmacokinetic behaviors. When adopting rational drug design, Lipinski’s rule of five is necessary to ensure the presence of drug-like features [21,33]. The four characteristics of Lipinski’s rule of five are followed by our synthesized compounds (mol. wt. ≤ 500 Da; log P o/w ≤ 5; HBD ≤ 5; HBA ≤ 10; and solubility (LogS): should be ≥4) [34,35,36,37]. Both of our synthesized compounds exhibit significant drug-like features because the ADME parameters for all of them were within the acceptable range, as shown in Table 2 and Table 3.

The estimation of the pharmacokinetic parameters of drug molecules enables researchers to predict some of their important biological aspects [38,39]. The optimal oral bioavailabilities of the compounds were predicted using Lipinski’s rule of five and Veber’s principles (Table 3). To evaluate the pharmacokinetic profiles and drug-likeness of both compounds, their ADMET properties were estimated (Table 2 and Table 3) [36]. The docking scores (kcal/mol), protein-ligand interactions, and other parameters, viz., active amino acid residues, bond lengths (A˚), and types of interactions of synthesized compounds one and two are shown in Table 4. Figure 3 and Figure 4 show all of the docked molecules in their respective 2D and 3D docking poses.

We designed and synthesized the potential lead compounds one and two through the SBDD approach based on the study of their affinity for the allosteric site of the target enzyme and calculated binding free energies. These designed compounds one, two, and the natural substrate LL-DAP displayed drug-likeness in accordance with Lipinski and Veber’s rules, and no violations of either rule were found in any case. The designed molecules one and two had calculated logP values of −0.33 and −0.99, respectively. This indicates good lipophilicity and high GI absorption. The compounds’ logP values demonstrate how readily these medicines enter their target tissues when administered to humans. Compounds one and two were both found to have two hydrogen bond donors (HBDs), while LL-DAP had four. Both compounds one and two had a molecular weight of 218.30 and 202.23 Da, respectively, and four and five hydrogen bond acceptors (HBAs), respectively, as required. Both of our compounds, as well as the native substrate LL-DAP, did not violate the Lipinski rule of five, which indicates their better absorption and lipophilicity. Both of these compounds are within the permissible range for oral availability and do not violate Veber’s criterion for total polar surface area (TPSA, which should be less than 140) and the number of rotatable bonds (which should be less than 10).

It was observed that our synthesized compounds did not show blood-brain barrier penetration potential, and thus they cannot be targeted for delivery to the central nervous system. These compounds had the best log *Kp* (skin permeability, cm/s) and bioavailability values. Both of these molecules did not violate the Ghose filter or the Egan rule but did violate the Muegge filter (Table 2). The molecules displayed high GI absorption and did not violate Lipinski and Veber’s rules. Subsequently, these compounds were then subjected to molecular docking investigations on *dap*F.

### 3.3. Molecular Docking Study

(a)Protein structure designing

The protein structure preparation (PDB ID: 5BNR) was performed using the protein preparation wizard within the software programme Maestro (version 11.5, implanted in Schrodinger) (made by Schrodinger.com in New York, USA). The pre-processing of proteins was conducted with the help of Prime, in which the addition of missing side chains was incorporated, and restrained minimization was also performed. The addition of hydrogen bond assignments was also performed. Finally, no problem was found in the protein structure. While the inner water molecules were preserved, the solvent water molecules near the 6-OCH_3_ location, as well as the waters beyond 5.0 A˚ from the ligand, were removed. The pH value was set to 7.0 ± 2.0, and Epik was used to produce the states of the ligands. The PROPKA program’s H-bond assignment was used to optimize the water molecules’ orientations, and the pH level was set to 7.0. The minimization of the energy of the added hydrogens was conducted via the OPLS3 force field. For the generation of the grid, the protein of compound three in the complex with *dap*F was utilized.

(b) Ligand structure design

The LigPrep programme with default parameters was used for the preparation of the ligands. An OPLS3 force field was used for energy minimization. A ligand docking module was used to carry out the docking studies. No constraints were applied, and all the parameters were left at their default values. The glide SP mode was used for the docking of the compounds.

The ligand-target docking was performed using the “Grip-Based Docking Tool” from the wizard within Maestro in Schrodinger for analyzing the structural complex of our target protein, viz., *dap*F, with LL-DAP and compounds one and two, for understanding the structural basis of this protein’s target specificity. The ligand and protein interactions were revealed by choosing the conformers from the file. The purified protein was opened, and grid generation was performed with a cubic box of specific dimensions centered on the protein cavity. Then, the ligand conformers were selected along with the folder simultaneously. The output folder was selected to save the results of docking. The protein-ligand interactions were investigated for the hydrophobic and hydrophilic properties of these complexes in order to understand the binding affinity of the ligand towards the protein. 

We chose compounds one and two for molecular docking and compared them with a natural substrate, LL-DAP. The docking studies suggested that compounds one and two interact more ardently with *dap*F than LL-DAP. The anticipated binding free energies (kcal/mol) were used to determine the molecular docking scores [40,41]. Each of our compounds’ docked positions was evaluated, and the pose with the lowest binding free energy was selected. Figure 3 shows the hydrogen bond interactions between compound one and the specific amino acid residues in the target protein, viz., *dap*F. The strongest ligand-protein affinity is shown by the best dock score, which has the lowest binding free energy. Compounds one and two had anticipated binding free energies of −9.823 and −10.098 kcal/mol with dapF, respectively, while LL-DAP, the natural substrate, had a binding free energy of −9.426 kcal/mol. This indicates that our synthesized molecules have a greater dock score and a more stable conformation with *dap*F than LL-DAP, thereby concluding their greater target-ligand affinity compared with LL-DAP.

(c) Docking with thio-DAP

The docking studies of compound one revealed good binding interactions with *dap*F compared to LL-DAP, which had a binding energy of −9.823 kcal/mol with *dap*F. Compound one formed eight hydrogen bonds with active site residues Asn74, Asn159, Asn194, Glu212, Arg213, and Thr223 at bond distances of 2.45, 0.81, 3.24, 1.82, 2.07, and 2.73 A˚, whereas LL-DAP formed twelve hydrogen bonds, as shown in Figure 3a. As shown in Table 4, the active site residues ASN74 and ARG213 formed two hydrogen bonds, one with the carbonyl oxygen and the other with the amine of the L-terminal of compound one, with short distances of 0.98, 3.24, 1.82, and 3.47 A˚, respectively. Figure 3a depicts a hydrophobic interaction between compound one and *dap*F that involves the active site residues Phe17, Tyr72, Ala80, Met82, Cys83, Pro160, Val215, and Cys221. The hydrophobic interaction of compound one with the active site residues (Cys83 and Cys221) of *dap*F is associated with the nitrogen of the thiazole ring of compound one at the site of epimerization on the enzyme’s catalytic site. The active site residues, Cys83 and Cys221, are the main catalytic residues involved in the epimerization of LL-DAP to meso-DAP, as shown in Figure 3a. The negatively charged residue of *dap*F Glu212 is involved in the hydrogen bonding and ionic interaction with NH_3_^+^ of the L-terminal of compound one. As shown in Figure 3, the polar active site residues of *dap*F ASN15, ASN74, ASN85, ASN159, ASN194, and THR223 are associated with an electronegative inductive attraction with the atoms nitrogen, sulphur, the two oxygens of thiazole, and the NH_3_^+^ group of compound one’s L-terminal. One of the positively charged active site residues of *dap*F viz., Arg213, is involved in the ionic interaction with the negatively charged carbonyl oxygen of the L-terminal of compound one, which is actually a salt bridge. The glide H-bond value was found to be −1.532, which indicates the formation of a stable hydrogen bonding network between *dap*F and compound one as compared to the *dap*F complex with LL-DAP. All the hydrophobic and hydrogen bonding interactions were observed in both terminals of compound one, viz., the epimerization terminal and L-terminal (the substrate recognition terminal).

(d) Docking with oxa-DAP

The binding free energy value obtained for compound two with *dap*F was −10.098 kcal/mol, which is slightly better compared to compound one (−9.823 kcal/mol) and LL-DAP (−9.426 kcal/mol). Thus, these values are lower than those shown by both compound one and LL-DAP. These docking studies suggest that compound two has a greater affinity and potency. Compound two establishes H-bonds with the amino acid residues Asn74, Asn159, Asn194, Arg213, Glu212, and Thr223 of *dap*F with the cumulative bond distances of 2.42, 0.78, 0.83, 2.26, 1.05, 2.83, 2.09, and 1.88 A˚, as presented in Figure 3b and Table 4. The active site residues of *dap*F, viz, Asn74 and Arg213, formed two hydrogen bonds, one with the carbonyl oxygen and another with the amine in the L-terminal of compound two, with the bond distances of 0.83, 2.26, 1.05, and 2.83 A˚, respectively, as provided in Table 4. Compounds one and two interact hydrophobically with *dap*F via the active site residues Phe17, Tyr72, Ala80, Met82, Cys83, Pro160, Val215, and Cys221 (Figure 3b). The hydrophobic interaction with the active site residues of *dap*F, viz., Cys83 and Cys221, is also associated with the nitrogen of the oxazole ring of compound two (similar to the case of compound one) at the site of epimerization on the catalytic site of this enzyme. The negatively charged residue of *dap*F, viz., Glu212, is involved in the hydrogen bonding and ionic interaction with the NH_3_^+^ L-terminal of both compounds (one and two). As shown in Figure 3a,b, some polar active site residues of *dap*F, namely Asn15, Asn74, Asn85, Asn159, Asn194, and Thr223, are associated with an electronegative inductive attraction with the atoms nitrogen, the oxygen of oxazole, and two oxygens and the NH_3_^+^ group of the L-terminal of both compounds one and two. Arg213 is involved in the ionic interaction with the carbonyl oxygen at the L-terminal of both compounds one and two (a salt bridge). The glide H-bond value was found to be −1.757 kcal/mol, which indicates the formation of a stable hydrogen bonding network between *dap*F and compound two as compared to the *dap*F complex with LL-DAP. All the hydrophobic and hydrogen bonding interactions were observed in both terminals of compound two, viz., the epimerization terminal and L-terminal (the substrate recognition terminal). The active site residues of *dap*F, viz., Cys83 and Cys221, are the main catalytic residues involved in epimerization, as represented in Figure 3b and Figure 5. The side chains of Cys83 and Cys221 were found to play an important role in the acid-base catalysis performed by this enzyme, *dap*F. *Dap*F catalyzes epimerization by employing a “two-base” mechanism. The stereoinversion involves two active site cysteine residues acting in concert as a base (thiolate) and an acid (thiol). During the design of compounds one and two, we replaced the central carbon atom of LL-DAP with a sulphur atom, which involves neither H-bonding nor hydrophobic interactions. This means that the isosteric replacement of the central carbon with sulphur, nitrogen, oxygen and other atoms did not affect the binding properties of the molecule at the active site.

(e) Docking with LL-DAP

The binding free energy of the natural substrate LL-DAP was found to be −9.426 kcal/mol with *dap*F, which is more than both compounds one (−9.823 kcal/mol) and two (−10.098 kcal/mol). The energy value of the conformers shows that LL-DAP has a higher docking score and less affinity than compounds one and two. LL-DAP was found to establish 12 H-bonds with the active site residues of *dap*F, viz., Asn15, Asn74, Gly84, Asn85, Glu212, Arg213, Asn159, Asn194, Gly222, and Thr223 with the bond distances of 1.34, 1.42, 4.22, 1.13, 4.53, 2.14, 1.21, 2.75, 2.23, 1.41, 3.14, and 3.31 A˚, respectively, while compounds one and two formed 8 H-bonds each as shown in Figure 3a–c. The active site residues of *dap*F, viz., Glu 212 and Arg213, formed two hydrogen bonds each, one with the carbonyl oxygen and another with the amine of the L-terminal of LL-DAP, with the bond distances of 1.13, 4.53, 1.42, and 4.22 A˚, respectively, as provided in Table 4. *Dap*F H-bonding residues Asn74, Asn159, Asn194, Glu212, Arg213, and Thr223, are common in hydrogen bonding scenarios with compounds one, two, and LL-DAP, though the bond lengths differ slightly. LL-DAP shows hydrophobic interactions with *dap*F that involve its active site residues, viz., Phe17, Tyr72, Ala80, Met82, Cys83, Val215, Val215, and Cys221 (Figure 3c). The amino acid residues of *dap*F, viz., Cys83 and Cys221, were found to be crucial in the epimerization and binding. These Cys83 and Cys221 residues govern the epimerization of LL-DAP. At a neutral pH, these amino acid residues exist as rapidly equilibrating thiolate-thiol pairs in the presence of the substrate. Both these residues viz., Cys83 and Cys221 of *dap*F appear in complex formation with compounds one, two, and LL-DAP. The negatively charged residue of *dap*F, viz., Glu212, is involved in the hydrogen bonding and ionic interactions with the NH_3_^+^ of the L-terminal of compounds one, two, and LL-DAP. Some polar active site residues of *dap*F, viz., Asn15, Asn74, Asn85, Asn159, Asn194, Thr218, Ser220, and Thr 223, are associated with an electronegative inductive attraction with two carboxylic groups and the NH_3_^+^ group of both the L-terminals of LL-DAP, as shown in Figure 3c. One of the positively charged active site residues of *dap*F, viz., Arg213, is involved in the ionic interactions with the carbonyl oxygen of the L-terminal of compounds one, two, and LL-DAP (a salt bridge). The glide H-bond value was found to be −1.455 kcal/mol, which indicates the formation of a stable hydrogen bonding network between *dap*F and LL-DAP. All the hydrophobic and hydrogen bonding interactions were observed in both the terminals of compound one, viz., the epimerization terminal and L-terminal (the substrate recognition terminal).

### 3.4. Antimicrobial Studies

As per the results of antibacterial investigations (Table 5), compounds one and two are effective against both gram-positive and gram-negative bacterial strains. When compared to the therapeutic agent ciprofloxacin, these compounds efficiently inhibited the development of the bacterial strains *E. coli, B. subtilis, P. aeruginosa,* and *B. megaterium* in vitro. This indicates that they have a broad antibacterial spectrum. These compounds have a bacteriostatic effect because their individual minimum inhibitory concentrations (MICs, µg/mL) prevent bacterial growth. These compounds were also evaluated for their bactericidal effects by determining their minimal bactericidal concentrations (MBCs, µg/mL). The results from the optical density growth readings at 24 h indicated that our synthesized compounds exhibited the same levels of MIC against the tested bacteria. A colorimeter was used to measure the turbidity at approximately 620 nm. Compounds one and two were shown to have good activity against *E. coli, B. subtilis, P. aeruginosa,* and *B. megaterium*, and their colorimetric analyses revealed that their MICs were 70 and 80 µg/mL, respectively. By monitoring the growth of microorganisms on culture media after 24 h at 37 °C, higher concentration solutions were utilized to determine the MBCs. For the bacterial strains indicated above, these substances demonstrated different levels of MBC. In tests against *B. subtilis*, *B. megaterium*, *E. coli*, and *P. aureogenosa*, compound one was considered to have excellent MBC values (MBC = 80 µg/mL, 90 µg/mL, 100 µg/mL, and 90 µg/mL, respectively). Compound two also exhibited approximately similar values of MBC against *B. subtilis, B. megaterium,* and *P. aureogenosa* (MBC = 90 µg/mL). By using the cup-plate agar diffusion method, the zone of inhibition (ZOI) of our synthetic DAP analogues and the standard antibacterial ciprofloxacin was examined in the culture media at concentrations of 100 µg/mL and 200 µg/mL. After 48 h, the activity was noticed on a shaking incubator at 37 °C. Better ZOI results for compounds one and two demonstrate their efficacy against the tested microorganisms. Figure 5 shows the graphs for these ZOI, which were measured in millimetres (mm) (Table 5). By isosterically substituting a carboxylic group with a thiazole and an oxazole moiety, compounds one and two were synthesized. At doses of 70–80 µg/mL, the compounds showed notable antibacterial effectiveness against all tested bacteria when compared to the standard antibiotic, ciprofloxacin. Compound one, with the thiazole moiety, in particular, showed somewhat more efficacy than compound two and exhibited the highest inhibition against *B. subtilis* and *B. megatorium*, where it was found to be nearly 187 times as potent as ciprofloxacin at both concentrations of ZOI. Therefore, these findings imply that the presence of thiazole and oxazole as carboxylic acid pharmacophores would be particularly significant in the process of developing new drugs. This could be attributed to their bulky nature, due to which they may impart higher lipophilicity, which in turn is helpful in their biological transportation and distribution. From the literature survey, it was found that the previously reported analogues were obtained with the replacement of both the carboxylic acid groups of DAP with phosphonic acid to produce the phosphonate analogue of DAP (P-DAP) [42,43]. However, these analogues were found to be resistant to unassisted crossing across cell membranes. Furthermore, because all types of bacteria use lysine biosynthesis as an important biochemical process in the DAP pathway, we predict that our analogues would be effective against resistant bacterial strains such as MRSA, VRE, and VRSA [34,44]. These resistant bacterial strains are the most distraught and virulent microorganisms that cause a broad array of problems for hospitalized patients and show multi-drug resistance to numerous currently available antibacterial agents.

### 3.5. Mechanism of Epimerization by dapF

Utilizing a “two-base” mechanism, the enzyme *dap*F converts LL-DAP (2) to meso-DAP (1). Two cysteine residues, Cys83 and Cys217, are involved in the reaction. In order to remove a proton from LL-DAP, the Cys83 thiolate acts as a base and removes the proton. In contrast, the Cys217 thiol re-protonates the molecule, creating a planar carbanionic intermediate (19) from the opposite side to generate meso-DAP (Figure 6). These cysteine residues appear as a quickly equilibrating thiolate-thiol pair in the presence of a substrate. Two active site cysteine residues work together as a base (thiolate) and an acid (thiol) to cause stereoinversion. The basicity is improved by eliminating hydrogen when a firmly bound thiolate-thiol pair is formed from the two cysteines at the active site without the use of a solvent [6].

### 3.6. SAR for Designing the DAP Analogues

Under normal conditions, the enzyme *dapF* epimerizes an amino acid’s conformation at the α-carbon without the use of any metal ions or cofactors. Proline racemase, aspartate racemase, glutamate racemase, and *dap*F are bacterial PLP-independent racemases that have recently had their crystal structures resolved. *Dap*F, however, stands apart from these other racemases because it distinguishes between two stereocenters [6]. To analyze the docking of the ligand-enzyme complex and identify the appropriate structural conditions for catalyzing epimerization, molecular docking and DFT simulations were used. Due to the H-bonding interactions that both the carboxylate and amino groups have with the active site residues, the molecular docking experiments demonstrated the significance of the distal carbon C-6 of LL-DAP in the formation of the complexes. The interactions between the functional groups bonded to the C-2 and the catalytic residues Cys83 and Cys217 of the binding cavity immobilize the ligand in a position suitable for the epimerization, according to DFT quantum mechanical estimations of the Michaelis complex and for specific interactions at the distal site, which only require the L-configuration. The experimental evidence further supported the hypothesis that the stereochemistry of the non-reacting (distal) C-6 carbon as the L terminal is crucial for ligand recognition and antibacterial activity. The enzyme distinguishes between these two stereocenters and interconverts LL-DAP to DL(meso)-DAP. Previous findings demonstrated that the aziridine and phosphonate analogues of DAP were synthesized using stereoselective methods in order to characterize the L-terminal as being essential for *dap*F inhibitors [45,46]. According to this, DAP analogues without an amino or carboxyl group in the L-configuration are neither effective substrates nor inhibitors [1].

The experimental evidence further supported the hypothesis that the stereochemistry of the non-reacting (distal) C-6 carbon as the L terminal is crucial for ligand recognition and antibacterial activity. The enzyme distinguishes between these two stereocenters and interconverts LL-DAP to DL (meso)-DAP. As a result, DAP analogues lacking an amino or carboxyl group in the L-configuration lack substrates and efficient inhibitors [1]. In vivo bioactive testing, the incorporation of various linkers (alkyl, aryl, and heterocyclic moieties), the use of various heterocyclic azoles (viz., benzimidazole, pyrazole, furan, thiophene, and benzotriazole), as well as the incorporation of various functional groups (viz., amide esters and ketones) may be introduced into the DAP backbone. However, the substitution of a heterocyclic ring in the DAP route results in exceptional enzyme inhibitory capabilities and amplifies the antibacterial effects of the recently produced analogues. The structural requirements for substrate recognition are fairly strict; the substrate must connect with the proper α-amino acid functionality at the active site and place the stereochemically correct polar carboxylic groups at the distal recognition site in order to serve as a substrate with efficient binding. *Dap*F from *E. coli* needs both carboxyl and amino groups in order to recognize the substrate and transform it (perform an α-hydrogen exchange) at the distal (non-reacting) α-amino acid site [1,14,47]. The pK_a_ of the neighbouring hydrogen may change significantly if the carboxyl group in the DAP is replaced with a heterocyclic ring. It is worth noting that the distal recognition sites of *dap*F prefer to bind to the natural α-amino acid moiety and require more structural fidelity than the protein groups that bind to the DAP molecule’s reacting terminus [14,47,48].

Furthermore, the antimicrobial potencies of compounds one and two are dependent on an arrangement resembling an amino acid at one of the substrate’s terminals, which restricts us from using L-cysteine as a precursor for the synthesis and necessitates the use of three to six carbon lengths of aliphatic chains from the amino acid’s distal recognition site [14]. DAP oxazole and imidazole analogues with different carbon lengths and an additional carboxylic group on the heterocyclic ring were reported [14]. The study demonstrated that activities might decrease as the aliphatic chain length increases or decreases. The isosteric substitution of DAP’s central carbon with sulphur, nitrogen, oxygen, and other atoms has no effect on the molecule’s binding properties at the active site. This carboxylic group to heterocycle transformation may modulate the lipid-to-water partition coefficient, affecting their diffusion in bacterial cells as well as their interactions with bacterial cells, thereby improving the pharmacokinetic properties.

## 4. Conclusions

The novel chemical analogues of DAP, viz., thio-DAP and oxa-DAP were synthesized using the SBDD approach. Both of these were found to be effective against gram-positive and gram-negative bacteria. Good values of MICs (70–80 µg/mL), MBCs (80 µg/mL–100 µg/mL), and ZOI were obtained when compared with ciprofloxacin. L-cysteine was used selectively in the synthesis of thio-DAP (1) and oxa-DAP (2). The antibacterial properties of compounds one and two were also estimated in silico with a molecular docking study. The hydrophobic and hydrophilic properties of these complexes were investigated using the programme version 11.5 Schrodinger based on the scoring functions. The docking scores of compounds one and two were −9.823 and −10.098 kcal/mol, as compared with LL-DAP (−9.426 kcal/mol). This suggests that compounds one and two interact more precisely with *dap*F than LL-DAP.

*In silico* toxicity risk assessments of the synthesized compounds were carried out using protox-II software. Both of these compounds were found to be nontoxic, as revealed by their sufficiently high LD_50_ values. Also, they were found to be inactive at nuclear receptors, i.e., the α-androgen receptor (AR), aromatase receptor, and estrogen receptor alpha (ER). Both these compounds demonstrated drug-likeness, and no violations of Lipinski’s and Veber’s rules were observed in their cases. Also, both had a log P value within the stated limit. Hence, these compounds have good lipophilicity and high GI absorption and could therefore be administered via the oral route. These compounds did not show BBB penetration potential, so they cannot be delivered to the central nervous system.

## 5. Chemistry

All starting materials and reagents were purchased from Dipa Chemical Laboratory, Aurangabad, and used without further purification. The solvents used were either of analytical grade or dried and distilled immediately prior to use. All the reactions were performed using oven-dried glassware. The melting points were recorded on a VEGO electrothermal digital melting point apparatus. Microwave-assisted organic synthesis (MAOS) was carried out in the scientific catalytic microwave system CATA-RI at 325 watts (50% power). Merck pre-coated Silica gel 60 F254 aluminium sheets (20 × 20 cm, layer thickness 0.2 mm) and Merck pre-coated Silica gel RP-C18 F254 aluminium sheets (20 × 20 cm, layer thickness 0.2 mm) were used for the TLC, and the spots were visualized with UV light (wavelength 254 nm) after being placed in an iodine chamber. All reaction products were kept in refrigerators at approximately 40 °C. The elemental analysis (% C, H, and N) was carried out with a Perkin-Elmer 2400 CHN analyzer. The IR (infrared) spectra of all compounds were recorded on a Prestige FT-IR spectrophotometer in KBr. The mass spectra were scanned on a Maldi TOF-MS spectrometer. All the analytical data has been provided in Appendix A.

### 5.1. Microwave-Assisted Synthesis of Thio-DAP (01) and Oxa-DAP (02)

K_2_CO_3_ (10 mmol) was added portion-wise to a solution of L-cysteine (2 mmol) in acetone (70 mL). The mixture was stirred for 0.5 h at the refluxing temperature. 2-bromo-methyl thiazole or 2-bromo-methyl oxazole (3 mmol) was then added to the solution, which was refluxed for 45 min in the scientific catalytic microwave system CATA-RI at the 325 watts (50% power). After the solvent was evaporated under a vacuum, the residue was chromatographed on a silica gel with petroleum ether or ethyl acetate as the eluent. An analytical sample was recrystallized from petroleum ether or ethyl acetate.

#### 5.1.1. Thio-DAP (01): (R)-3-((thiazol-2-yl)methylthio)-2-aminopropanoic Acid

White crystalline solid; molecular weight 218.3 gm/mol; molecular formula C_7_H_10_N_2_O_2_S_2_; % yield-69.31; m.p.: 192–194 °C; Rf value: 0.71. Anal. Calcd. For C_7_H_10_N_2_O_2_S_2_: C (38.51%), H (4.62%), N (12.83%), O (14.66%), S (29.38%). FTIR (KBr disk) 3061.03, 2544.11 (N-H stretch (primary amine)), 3343.50 (O-H aromatic stretch), 2866.22 (C-H alkanes stretch), 2083.12 (O-C-N), 1743.32 (C=O carboxylic acid), 1649.14 (>C=N- stretch), 1517.98 (>C=C< (aromatic conjugation)), 1425.40, 1352.10 (methyl (-CH_3_) bending), 1300.00 (N-C-S thiazole ring stretch), 1195.87 (C-O), 1064.71 (C-N amine), 1001.06 (N-C-S cyclic, thiazole ring), 943.19 (alkene (out of plane bend)), 869.90, 821.68, 771.53 (aromatic (out of plane bend)), cm^−1^. ^1^H NMR (CCl_4_-d_6_ 400 MHz) δ ppm: 2.639, 3.182 (d, methylene proton), 3.764, 3.849, 3.912 (t, methylene protons), 5.288 (s, -NH_2_ protons), 7.481, 7.588 (d, methylene protons of thiazole), 11.121 (s, -OH). ^13^H NMR (CCl_4_-d_6_ 400 MHz) δ ppm: 35.891, 36.227, 55.892, 119.271, 142.768, 166.892, 170.281, 172.992, 173.992. Maldi TOF-MS: *m/z* values, 100.125 (100.0%), 130.583 (36.54%), 136.054 (19.49%), 220.019 (33.58.35%) (M + 1).

#### 5.1.2. Oxa-DAP (02): (R)-3-((oxazol-2-yl)methylthio)-2-aminopropanoic Acid

White crystalline solid; molecular weight- 202.23 gm/mol; molecular formula- C_7_H_10_N_2_O_3_S; % yield-73.32; m.p.: 215–217 ºC; Rf value: 0.61. Anal. Calcd. For C_7_H_10_N_2_O_3_S: C (41.57%), H (4.98%), N (13.85%), O (23.73%), S (15.86%). FTIR (KBr disk) 3061.03, 2544.11 (N-H stretch (primary amine)), 3343.50 (O-H aromatic stretch), 2866.22 (C-H alkanes stretch), 2083.12 (O-C-N), 1743.32 (C=O carboxylic acid), 1649.14 (>C=N- stretch), 1517.98 (>C=C< (aromatic conjugation)), 1425.40, 1352.10 (methyl (-CH_3)_ bending), 1300.00 (N-C-O oxazole ring stretch), 1195.87 (C-O), 1064.71 (C-N amine), 1001.06 (N-C-O cyclic, oxazole ring), 943.19 (alkene (out of plane bend)), 869.90, 821.68, 771.53 (aromatic (out of plane bend)) cm^−1^. ^1^H NMR (CCl_4_-d_6_ 400 MHz) δ ppm: 2.732, 3.272 (d, methylene proton), 3.665, 3.740, 3.817 (t, methylene protons), 5.389 (s, -NH_2_ protons), 7.582, 7.687 (d, methylene protons of thiazole), 11.241 (s, -OH). ^13^H NMR (CCl_4_-d_6_ 400 MHz) δ ppm: 35.791, 36.328, 55.7923 119.381, 142.898, 166.991, 170.274, 172.902, 173.972. Maldi TOF-MS: *m/z* values, 83.735 (100.0%), 115.535 (20.15%), 159.325 (14.57%), 203.121 (21.03.35%).

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
