# Peer review of "The Design, Synthesis, and Evaluation of Diaminopimelic Acid Derivatives as Potential dapF Inhibitors Preventing Lysine Biosynthesis for Antibacterial Activity"

_antibiotics, 2022, doi:10.3390/antibiotics12010047_

Round 1

Reviewer 1 Report

Dear Editor, dear Authors, Mohd Sayeed Shaikh et al., submitted a paper on the synthesis of Thiazole and oxazole analogues of Diaminopimelic Acid (DAP) by replacing it’s carboxyl groups and the central carbon atom was replaced by Sulphur using structure-based drug design (SBDD) technique. The authors aimed in their study to increase inhibitor binding affinity and cellular potency towards the target site by using SBDD. They have performed toxicity, ADME, molecular docking and antibacterial properties studies on the synthesized derivatives. For my opinion, the manuscript is of interest, however it is not well written. I suggest that the authors should revise their manuscript, especially the materials and methods part, where they should include all details about the methods in an appropriate manner for reproducibility of the work. Therefore, I believe that the manuscript cannot be accepted in its present format for publication in Antibiotics MDPI Journal. I have the below comment to the authors that should be revised.

Comments to the authors

-          Page 3 Materials and methods. This part in the manuscript is not written as usual; the authors here should not discuss other methods from the literature, and they should describe only the chemicals, methods and techniques used for characterization of their products as well as methods appropriate to their work.

Sincerely yours,

Author Response

The detailed response to the Reviewer's comments and Suggestions has been attached herewith

Reviewer 2 Report

Shaikh et al reported the molecular docking of diaminopimelic acid derivatives as dapF Inhibitors and tested the antibacterial activity. Comments on this manuscript are shown below.

1.  Line 248: What is the rationale of the statement Log P -0.33 and -0.99 indicating good lipophilicity and high GI absorption.

2. In Table 1, BBB permeability, P-gp substrate, and CYP enzyme inhibitors should be reported.

3. Table 3, the toxicity results should be included.

4. How did the authors prove mechanisms of epimerization?

5. In Table5, the values should be average and SD and the statistical analysis should be applied to compare the results.

Author Response

The detailed response to Reviewer-2 comments and Suggestions has been attached herewith

Reviewer 3 Report

In the present manuscript, authors synthesized thiazole and oxazole analogs of DAI as potential dapF inhibitors and tested these analogs against their antibacterial activity. The authors also performed in silico ADME, Toxicity, and docking study. This is interesting; however, it needs significant revision to improve the quality of the manuscript. Here are my comments: 

1.     Since the authors did not perform in vitro dapF activity; therefore, the title should include the word “potential” before dapF inhibitors.

2.     This manuscript needs significant English editing. Sometimes, it is hard to understand what the authors are trying to explain, and it needs refinement in the total manuscript contents.

3.     Abstract: “Thiazole” should be written as “thiazole.” Define “DAI.”

a.     “Good values of MIC, MBC.” What are good values? Please refine the sentence.

b.     “Dock score” should be written as “docking score”.

4.     Material and Methods:

a.     This section should be written in a concise format. The authors can move some parts to the discussion section.

b.     Scheme 3: Synthesis of Oxa-DAP (3) should be written as Synthesis of Oxa-DAP (2). Please check the entire manuscript for consistency. Stereodescriptor is missing for compounds 8 and 2. 

5.     Pages 8-10; section 3.3. Molecular Docking Study: This section should be written in a conservative style. Please remove the unnecessary repetition of sentences or wordings.

a.     The authors did not mention the PDB id for protein in the entire manuscript.

b.     “Grip Based docking tool” or “Grid-Based docking tool”

c.     Page 9, lines 298-299: “The purified protein”???

d.     Page 9, Lines 307-308: The authors only calculated the GlideScore/Docking score, not binding free energy; please correct this statement.

e.     The amino acid numbering should be presented in ascending order.

6.     Page 17: Lines 602 and 618: The stereodescriptor “R” should be italicized.

7.      The authors should use a real minus sign when they report docking scores.

8.     “µg/ml” should be written as “µg/mL” in the entire manuscript. Please check Table 5, Figure 5.

9.     What is the Y-axis for Figure 5?

10.  Please verify the Table 5 explanation on Page 13.

11. It would be better if authors can provide 3D overlay figures of compounds 1 and 2 at the active site of the protein. 

Author Response

The detailed response to Reviewer-3 comments and Suggestions has been attached herewith

Round 2

Reviewer 1 Report

Dear Editor, dear Authors, Mohd Sayeed Shaikh et al., submitted a revised manuscript of their paper entitled “Design, Synthesis and Evaluation of Diaminopimelic Acid Derivatives as dapF Inhibitors Preventing Lysine Biosynthesis for Antibacterial Activity” submitted to Antibiotics MDPI Journal. For my opinion, the authors answered all the reviewers’ comments and as the revised manuscript is now much better written. Therefore, I believe that the manuscript can be accepted in its present format for publication in Antibiotics MDPI Journal.

Sincerely yours,

Reviewer 2 Report

The manuscript is acceptable for publication.